# Impact of Precipitation and Temperature Variability of the East Asian Summer Monsoon (EASM) on Annual Radial Increment of Selected Tree Species in Northeast China

**Sandra-Maria Hipler [1,\*], Benedikt Speicher [1], Lars Sprengel [1] , Hans-Peter Kahle [1] , Heinrich Spiecker [1] and Shuirong Wu [2]**

[1] Chair of Forest Growth and Dendroecology, Albert-Ludwigs-University Freiburg, Tennenbacher Str. 4, 79106 Freiburg, Germany; benedikt.speicher@yahoo.de (B.S.); lars.sprengel@iww.uni-freiburg.de (L.S.); hans-peter.kahle@iww.uni-freiburg.de (H.-P.K.); instww@uni-freiburg.de (H.S.)

[2] Research Institute of Forest Policy and Information, Chinese Academy of Forestry, No. 1 Dongxiaofu, Haidian District, Beijing 100091, China; shuirongwu@126.com

\* Correspondence: sandra.hipler@iww.uni-freiburg.de; Tel.: +49-761-2033737

**Abstract:** A dendroclimatological approach was used to analyze growth responses of the tree species *Pinus tabuliformis* Carr., *Larix gmelinii* Rupr., *Picea asperata* Mast. and *Quercus mongolica* Fisch. ex Ledeb. in a region of temperate climate in Northeast China. Annual radial increment (ARI) measurements from stem cross-sections were used to identify the effects of precipitation, air temperature and standardized precipitation evapotranspiration index (SPEI) on tree growth under monsoon-related conditions. We analyzed the ARI of 144 trees from 49 forest stands and applied response function and moving correlation analysis as well as a linear mixed-effects model to detect climate signal in the tree-ring series. Analyses of climate-growth relations confirmed the influence of monsoon intensities on ARI, especially in the months of May to July of the current year. Particularly in times of a weak monsoon, the preceding autumn months significantly affect the ARI. The positive effect of precipitation in times of a strong monsoon and the negative effect of air temperature-indicating increased evapotranspiration-in times of a weak monsoon alternate. An increase in drought sensitivity of the ARI was found, especially after long dry periods. The results revealed for *L. gmelinii* the highest climate sensitivity, with ARI more strongly influenced by precipitation in the monsoon-related months, whereas *Q. mongolica* was most drought tolerant and recovered quicker after growth depression. *P. asperata* and *P. tabuliformis* were located in between. Our findings provide evidence for a strong influence of the periodically fluctuating monsoon intensities on the ARI of all investigated tree species. Our results support decision-making for forest management under anticipated climate change, especially for tree species selection, in the climate sensitive region of Northeast China.

**Keywords:** annual radial increment (ARI); climate-growth relation; climate sensitivity; East Asian Summer Monsoon (EASM); Standardized Precipitation Evapotranspiration Index (SPEI)

## 1. Introduction

Considering climate change, adapting current and future tree species composition in the climate-sensitive region of Northeast China with semi-arid to arid climatic conditions is a challenge. Scarce scientific results on the effect of climate on the annual radial increment (ARI) of various tree species furthermore limits qualified decision-making concerning choice of tree species and tree species composition, and the assessment of species-specific responses to climate change. Climatic effects on

ARI in Northeast China have not been researched sufficiently, despite the fact that such knowledge is essential to assess species-specific tree growth responses to anticipated climate change and the effects of climate change upon forest productivity. The comparison of radial increment chronologies and meteorological time-series provides understanding of the interplay between terrestrial ecosystems and external forces reaching far back in time [1,2].

Reliable meteorological records have existed in China for about 60 years. Therefore, development of tree-ring chronologies became one of the most important tools in the study of past climate until the 1980s, due to increasing tree-ring sampling and applying special software for tree-ring analysis [3]. Nowadays, tree-ring chronologies of living trees, but also of dead or sub-fossil trees, can help to reconstruct temperature, precipitation and cloud cover that extend back several centuries to millennia. Shao et al. [4], for example, developed a 3585 years tree-ring width chronology of *Juniperus przewalskii* Kom. in the northeastern Qinghai-Tibetan Plateau composed of archaeological wood samples and samples of living trees. So far, dendrochronological studies are concentrated in the cold and arid regions of West and Northeast China, while in warm and humid regions they are relatively scarce, due to the fact that tree-rings in these regions may not be as sensitive to climatic factors [5]. Another reason is that not all species in warm-humid climates necessarily produce annual growth rings. In the context of global and regional warming, the duration of hot waves has been significantly extended [6]. The precipitation in northern China is decreasing while precipitation intensity is increasing and snow melting in spring starts earlier. All these changes lead, on the one hand, to an increase in the length of the growing season of coniferous forests at an average 3.9 days per decade and, on the other hand, to an alternation of species distribution [5,7]. Regional warming makes temperate broad-leaved forests more competitive and leads to a decrease in the proportion of boreal coniferous forests in Northeast China [7]. It is also expected that the abundance of trees growing in high altitudes and in middle and high latitudes will increase, while trees growing in the southern margin will suffer from high temperature and extreme drought [6]. The warm temperate zone and temperate zone in northeast China will be enlarged and the forests in the mountains will migrate to higher elevations [8].

The study area was chosen to best represent climate-growth relationships under different moisture conditions, in a region where tree growth is influenced by a complex interplay of various climatic parameters as well as by biotic and abiotic disturbances and competition for water [9,10]. The weather conditions in Northeast China are characterized by the influence of the East Asian Summer Monsoon (EASM) and thus periodically either cool and wet (strong monsoon) or warm and dry (weak monsoon) weather is prevailing [9]. Recent studies on climate reconstructions in Northeast China confirm these periodically fluctuating weather conditions [9,11–14]. Therefore, a significant influence of the East Asian Summer Monsoon (EASM) on the macroclimate in Northeast China can be assumed. Such reconstructions could help to better anticipate possible responses to future climatic changes as well as to better understand those of the past.

Additionally, Liu et al. [15] revealed a high synchronicity among tree-ring chronologies of *Pinus tabuliformis* Carr. with periodic climatic fluctuations. Analyses of climate-growth relationships in the study area confirm the great influence of monsoon intensities on ARI, especially in the monsoon-affected months of May to July of the current year and the preceding autumn. Water availability is crucial for the vitality of trees and composition of forest communities. Dendroclimatological studies verify that weather conditions during the vegetation period have a strong effect on the current growth of a tree, but also on the accumulation of non-structural carbohydrate storage for subsequent growth [15].

In this study, we focus on the relationship between the ARI of four native tree species and the climate variables precipitation, air temperature and standardized precipitation evapotranspiration index (SPEI) in the research area, using multivariate statistics. The hypotheses tested were:

1. The periodically fluctuating monsoon intensities in the research area have a significant effect on the ARI of the four investigated tree species.
2. The four tree species show significant differences in terms of their climate sensitivity.

## 2. Materials and Methods

### 2.1. Characteristics of the Study Site

The study area is located in the forest farms of Mulan Weichang state-owned Forest Farm Administration Bureau (hereinafter referred to as Mulan Forest) in Weichang County of Hebei Province in Northeast China (N 41°35′–42°40′, E 116°32′–117°94′; elevation: 950–1750 m above sea level). The predominant forest type in this area is a temperate coniferous forest dominated by *Pinus tabuliformis* [16]. The sample trees for this study were selected in planted, even-aged and well-stocked single-species stands of the indigenous tree species *P. tabuliformis*, *L. gmelinii* Rupr., *P. asperata* Mast. and *Q. mongolica* Fisch. ex Ledeb.

The climate of Hebei province is characterized as humid continental, with a warm and wet summer and a cold and dry winter [9]. It is dominated by the EASM, a subtropical monsoon affected by El Nino-Southern Oscillation (ENSO), which carries moist air from the Indian Ocean and Pacific Ocean to East Asia [17,18]. The climate data used in this study are from the meteorological station near the town of Weichang County at 844 m above sea level (meteorological station of Weichang County, Hebei province, 2013). Since baseline climate usually refers to 30 year periods, the monitoring period 1925–2013 is split into 1925–1950, 1951–1980 and 1981–2013 to present the respective figures (Table 1). The mean annual precipitation totals in this region range from 434 mm to 456 mm with a minimum of 237 mm (1951) and a maximum of 684 mm (1959). Due to the EASM, approximately 70% of the average 443 mm of annual precipitation occurs from July to August with a peak of 138 mm in July. Areas at 2000–2500 m above sea level have snowfall every year that typically lasts for about 4 months. For the three periods, the mean annual air temperature ranges from 4.6 °C to 5.5 °C with a maximum mean monthly air temperature of 21.3 °C in July and a minimum of −13.9 °C in January. The frost-free period encompasses 128 to 210 days and on average 169 days per year.

**Table 1.** Mean monthly precipitation sum, mean monthly air temperature, mean annual precipitation totals (MAPT) and mean annual air temperature (MAAT) at the meteorological station of Weichang County, Hebei province, 2013. The monitoring period is split into the sub-periods 1925–1950, 1951–1980 and 1981–2013.

| Months | Precipitation [mm] | | | Air temperature [°C] | | |
|---|---|---|---|---|---|---|
| | 1925–1950 | 1951–1980 | 1981–2013 | 1925–1950 | 1951–1980 | 1981–2013 |
| Jan | 1 | 2 | 1 | −13.9 | −13.2 | −12.5 |
| Feb | 2 | 4 | 3 | −10.5 | −10.3 | −8.6 |
| Mar | 7 | 8 | 7 | −3.1 | −2.8 | −1.4 |
| Apr | 15 | 15 | 18 | 6.7 | 6.4 | 7.5 |
| May | 43 | 37 | 38 | 13.5 | 13.8 | 14.6 |
| Jun | 63 | 75 | 74 | 18.5 | 18.1 | 18.9 |
| Jul | 136 | 138 | 122 | 21.3 | 20.7 | 21.3 |
| Aug | 99 | 103 | 91 | 19.3 | 18.9 | 19.7 |
| Sep | 47 | 48 | 50 | 13.4 | 12.8 | 13.8 |
| Oct | 17 | 21 | 22 | 6.3 | 5.9 | 6.5 |
| Nov | 6 | 5 | 6 | −3.7 | −3.3 | −3.2 |
| Dec | 2 | 2 | 2 | −12.0 | −11.0 | −10.3 |
| MAPT | 438 | 456 | 434 | | | |
| MAAT | | | | 4.6 | 4.7 | 5.5 |

### 2.2. Field and Laboratory Measurements

In order to exclude inter-specific competition, even-aged monocultures and well-stocked stands with relatively low-intensity management regimes were preferably selected. Since growth data of mature trees provide relevant information about long-term tree growth, the selected forest stands were as old as possible to maximize the observation period.

Exclusively dominant trees according to Kraft classes one and two [19] were selected for stem disc collection, as these trees experienced less influence from competitors and silvicultural measures on their growth. The sample trees were chosen in closed canopy surroundings, in order to reduce effects of gaps, skid trails or admixed tree species in the direct neighborhood. To be included in our study, sample trees were required to have a healthy crown and no deterioration by felling or skidding damage to determine the ARI at 1.3 m tree height. The numbers of stands and trees for stem analysis are summarized in Table 2. Sampling took place during the growing season of 2014.

**Table 2.** Number of stands and trees for stem analysis for each tree species and figures for tree age, tree height, diameter at breast height (DBH), ARI and number of annual tree-rings at 1.3 m above ground.

| Parameter | | Tree Species | | | |
|---|---|---|---|---|---|
| | | *L. gmelinii* | *P. tabuliformis* | *P. asperata* | *Q. mongolica* |
| Number of stands | | 10 | 10 | 10 | 10 |
| Number of trees | | 28 | 30 | 29 | 30 |
| Tree age (y) | Range | 39–98 | 40–106 | 27–106 | 40–102 |
| | Mean | 67 | 76 | 54 | 68 |
| | SD | 20.4 | 23.4 | 29.7 | 9.6 |
| Tree height (m) | Range | 9.0–30.6 | 10.0–25.8 | 10.8–27.7 | 7.4–16.8 |
| | Mean | 21.5 | 17.8 | 18.5 | 12.7 |
| | SD | 7.3 | 4.9 | 5.2 | 2.7 |
| DBH (cm) | Range | 13.6–58.0 | 21.2–46.8 | 18.3–42.9 | 15.0–29.2 |
| | Mean | 35.5 | 34.3 | 27.9 | 24.9 |
| | SD | 11.8 | 7.0 | 7.6 | 12.3 |
| ARI at 1.3 m (mm) | Range | 0.2–8.7 | 0.2–6.8 | 0.2–8.3 | 0.3–4.6 |
| | Mean | 2.4 | 2.1 | 2.9 | 1.4 |
| | SD | 1.4 | 1.1 | 1.5 | 0.7 |
| Number of annual tree-rings at 1.3 m | Range | 37–96 | 36–103 | 24–99 | 34–96 |
| | Mean | 63 | 71 | 47 | 66 |
| | SD | 20.4 | 22.4 | 27.1 | 10.1 |

The tree height was assessed using a digital clinometer (Forester Vertex type III) and DBH using a diameter tape. The selected trees were felled and sectioned following standard procedures outlined in Van Laar and Akça [20].

For investigation of ARI, the stem disks were first air-dried, sanded with an orbital sander and then scanned (ScanMaker 9800 XLplus, Microtek). For measuring and crossdating the tree-rings, the scanned images were imported in "WinDendro[TM]" [21]. ARI was calculated from the quadratic mean value of eight radii per stem disc. Mean radial increment chronologies were created from the measurements by merging different series between trees of the same species, using the arithmetic mean. For each tree species, a separate chronology was established.

### 2.3. Statistical Analysis Methods

All statistical methods and data analyses were performed in the R programming environment version 4.0.0 using the Graphical User Interface R Studio and the statistical add-in XLSTAT version 2020.1 for Microsoft Excel 2016 [22–24].

### 2.3.1. Exploratory Statistics

Prior to the actual analysis two dendrochronological indicators, the Gini-coefficient (G) and the Expressed Population Signal (EPS) are used to describe the measured ARI series and to check their suitability for the intended analysis. Biondi and Qeadan [25] use G to describe the whole heterogeneity of an examined ARI series. Compared to the mean sensitivity, G specifies the immediate adjacent

values in an ARI series and considers their heterogeneity over the whole observation period. Therefore, the Gini-coefficient is used in this study to quantify the heterogeneity of the examined ARI series. EPS provides an estimate of how closely a mean tree-ring chronology expresses its hypothetically perfect chronology based on an infinite number of trees. Wigley et al. [26] defined a threshold fixed by value EPS > 0.85 for the examination unit to reflect the whole population and to perform accurate climate–growth analyses. The calculation of the EPS is based on the correlation coefficient between the tree-ring series and the number of observed trees following the standard procedure outlined in Briffa and Jones [27]. The mentioned ratios are calculated, based on the detrended ARI series of the individual trees, using the R-package *dplR* [28]. Buras [29] critically assessed this threshold value, because there is no obvious way to determine how high the EPS should be to determine whether a tree-ring chronology is suitable for climate reconstruction purposes, since the strength of the common signal cannot be interpreted solely in climatic terms. He concluded that the evaluation of the reconstruction ability of tree-ring chronologies has to be based as well on other metrics such as bootstrapped response coefficients (see Section 2.3.4).

In order to investigate the effect of climate on ARI, monthly meteorological records from Weichang climate station for the period 1951–2013 were used. The time series data were verified for homogeneity to produce regional composite meteorological series covering the years from 1951 to 2013. The homogeneity was tested, using Levene's Test [30], comparing gridded datasets (N 41°–43° to E 116°–118°) of monthly air temperature and precipitation sum (Climate Research Unit (CRU)) with the measurement data of the meteorological station. The overall 1512 grid cells showed a close relationship between the monthly precipitation sum of the meteorological station Weichang and the gridded dataset. Hence, the climate data of Weichang were considered as suitable predictor variables for climate–growth analyses. The test statistics of the Levene's Test are shown in Appendix A Table A1.

### 2.3.2. Climate Data Analysis

Since most of the trees reached an age over 60 years, the weather data were extrapolated to the year 1925 based on the method described in Paesler [31]. The gridded data and the data of Weichang meteorological station are well correlated and the differences of monthly mean air temperature as well as the quotient of monthly precipitation sums are nearly constant. The following linear regression holds,

$$y_i = a + b * x_i + \varepsilon_i, \tag{1}$$

where $x_i$ represents the series of climate data (mean monthly air temperature and monthly precipitation sums) in year $i$ of Weichang and $y_i$ the series of climate data of the gridded datasets. *A* and *b* are constants, $\varepsilon_i$ is the residual error term. The mean of the *n* observed values was generated by:

$$\sum_{i=1}^{n} y_i = n * a + b * \sum_{i=1}^{n} x_i, \tag{2}$$

$$\overline{y} = a + b * \overline{x}, \tag{3}$$

For a series of mean air temperature values, $b = 1$, hence the expected constant difference from *a* can be described by $a = \overline{y} - \overline{x}$. In the case of precipitation, the constant $a = 0$, because the straight line of measurements passes through the origin. With $b = \frac{\overline{y}}{\overline{x}}$, one obtains the constant average quotient.

To include the effects of precipitation and air temperature variability on drought conditions, a simple multiscalar drought index-SPEI-was calculated [32,33]. The SPEI is a statistical indicator, based on the climatic water balance (monthly difference between precipitation and potential evapotranspiration, PET) and displays dry and wet periods. The use of SPEI offers another way of mitigating the effects of predictor multicollinearity (see Section 2.3.4) by aggregating the two negatively correlated parameters air temperature and precipitation. For the calculation, PET was determined from meteorological parameters measured at the Weichang climate station as well as the latitude of the research area to derive a monthly drought index, following the method outlined in

Vicente-Serrano et al. [32]. The SPEI at 1-month lags (SPEI-1) was calculated from monthly precipitation sums and monthly air temperature data during the observation period, using the R-package *SPEI*, version 1.7 [34].

### 2.3.3. Growth Data Analysis

The extraction of climate signal from tree-rings is hampered by a variety of factors, which arise from the complexity of tree growth. In this context, detrending by, for example, the use of smoothing splines based on low-pass filters, is a necessary procedure to extract climate signal from other non-climatic influences [35]. As the ARI series, used in this study, are derived from stands with different densities, the method of Cook and Peters [35] is appropriate for standardizing forest interior ARI series. The analysis of climate data-especially in the 1950s and 1960s—showed significant periodic fluctuations in weather conditions influenced by the EASM. The observation of these fluctuations could be confirmed by other research findings in Northeast China [9,11–14]. Liu et al. [15] revealed a high synchronicity in the ARI chronologies of all investigated tree species with periodic climatic fluctuations. In order to preserve this synchronicity, the measurement series of all investigated tree species have been detrended by fitting a spline with a 50% frequency cut-off of 60 years. Finally, to form the index series, the original values of the ARI series were divided by the values of the smoothing function. For monthly regression analysis, detrending was necessary in order to remove the non-climatic effects. All calculations were carried out with the *dplR* package of Bunn [28].

To assess the strength of the climate signal in the detrended ARI chronologies of the four tree species, a principal components analysis (PCA) was performed using the statistical add-in XLSTAT for Microsoft Excel [24]. The principal components can provide information about tree species-specific differences in ARI in terms of their intensity and direction.

### 2.3.4. Analysis of Climate-Growth Relations

To specify climate-growth relationships, orthogonalized climatic variables in regression were used, since they are an accepted method in dendroclimatic studies to analyze various climatic influences on tree-ring growth [36–38]. A general problem in multiple response function analysis could emerge from multicollinearity of climate factors, which leads to imprecise and unstable estimates of regression coefficients and their statistical significance. Hence, we applied the response function analysis of Fritts et al. [39], to overcome the problem of multicollinearity [40]. The predictors are first orthogonalized along principal components using the PVP criterion of Guiot [41] and the principal component with the lowest intrinsic values eliminated from the model [42]. Principal component regression under moderate multicollinearity still shows competitive performance [43] and is not necessarily the best approach to deal with multicollinearity. Guiot [41] proposed the bootstrap method of Efron and Tibshirani [44] to test for significant regression as one solution to obtain more robust parameter estimates. The inclusion of high-order eigenvectors in the response function is also possible, but can either increase or decrease its fidelity and should be compared with correlation functions for the decision on whether this inclusion was justified [37]. Blasing et al. [37], however, pointed out that statistically significant relationships between monthly climatic variables and tree growth may occur more often in response function elements than in correlation functions because of differences in the validity of the significance tests. The bootstrapped response function was done using the R package *treeclim* [45]. The calibration window was set from August of the previous year to October of the current year. For all results of the bootstrapped response function analysis, the $\alpha$-level was set to 0.05.

For evaluating the stationarity of dendroclimatic responses, moving bootstrapped correlation functions of Biondi [38] were calculated with the Gershunov test implemented in the function *dcc* of the R package *treeclim* [45]. To test for temporal instability of climate–growth relations, for example, the divergence phenomenon [46], a time-split observation of the climate signal is necessary to display the periodic fluctuations of precipitation and air temperature in Northeast China [9,11–13]. The analysis

of two decades can provide information about the varying response of tree growth to climate in terms of their intensity and their alignment over time.

In order to test for differences in the growth response between the analyzed tree species, the relationship between ARI, precipitation sum in the vegetation period (precv) and tree species (TS) was analyzed with an analysis of covariance (ANCOVA), using the *lmerTest* package [47]. The ANCOVA represents a method of comparing regression slopes between groups to assess different growth patterns [48]. Thus, the regression between precv and ARI may vary between tree species if they grow differently. To examine differences in sensitivity to precv between the tree species, regression lines are compared by studying the interaction of the two independent variables TS and precv. A random effect for the individual tree is added to characterize idiosyncratic variation that is due to individual differences. The formula is as follows:

$$\log(\text{ARI}) \sim \text{TS} + \text{precv} + \text{TS} * \text{precv} + (1|\text{ID}) \tag{4}$$

If the regression lines have the same slope but different intercepts, the sensitivity is similar for all species, but one species simply grows more slowly than the other. An interaction significantly different from zero indicates different slopes of the regression lines and is interpreted as a significant change in sensitivity rates among the tree species. The steeper the slope of the regression line, the greater the sensitivity [48]. To reduce the effect of heteroscedasticity, the over-estimation of the significance of the model, the dependent variable ARI was log-scaled to obtain more normal residuals. Since precv is plotted ascending on the *X*-axis, the years of measurement are mixed, which prevents the risk of overlay by age trend in the analyses. In order to investigate a long period of time, exclusively trees older than 50 years were compared. To fit a linear mixed-effects model, the package *lme4* [49] was applied, and for visualization of the interaction with the independent variables, an effect plot was produced based on the package *effects* [50].

## 3. Results

### 3.1. Characterization of Annual Radial Increment and Precipitation during the Vegetation Period

The Gini-coefficients in Table 3 show a similar heterogeneity for the investigated ARI series over all tree species throughout the observation period. Furthermore, intra-series EPS values for all tree species were over 0.85, indicating that the theoretical population is well represented. According to the definition of Wigley et al. [26], the ARI series can be used as a basis for climate–growth analysis.

**Table 3.** Gini-coefficient (G) and Expressed Population Signal (EPS) for the ARI series of the four investigated tree species.

| Indicator | *P. asperata* | *P. tabuliformis* | *L. gmelinii* | *Q. mongolica* |
|:---:|:---:|:---:|:---:|:---:|
| G | 0.615 | 0.571 | 0.581 | 0.602 |
| EPS | 0.865 | 0.908 | 0.904 | 0.884 |

The detrended ARI chronologies of the four investigated tree species and precv for the time span 1925 to 2013 are displayed in Figure 1. One striking feature is the strong synchronicity of the ARI of all four tree species with precv (variance 68.3%, Appendix B Table A2). The effect of growth depression is greater, the lower the precipitation sums are during preceding years. This is revealed particularly clearly in the dry years 1927, 1935, 1940, 1951, 1961, 1963, 1968, 1984, 2000 and 2007, which are characterized by extremely low precipitation sums between 237 mm and 395 mm in the vegetation period (April to October) and led to growth depression of all four tree species. Vice versa, all species show high increments during successive wet periods.

The year 1951 represents the most extreme dry year within the analyzed period. With a precipitation sum of 224 mm during the vegetation period, the sum is around 47% lower than the long-term average of 443 mm. A strong growth depression is characteristic for all four tree species in this year. Extremely low precipitation sums during the summer months of 1961–1963 led to drought conditions in the following vegetation period. In the preceding autumn months and in the months of current May to July 1959/1960, the trees received twice as much water as in 1960/1961. Since the trees may still have benefited from the high precipitation sums in 1959, which show the highest values for the autumn months of August to October in the whole observation period, reduced growth is less pronounced in 1960. During several relative wet vegetation periods, such as 1945–1950, 1953–1959, 1974–1979 or 1990–1994, the ARI was relatively high as well (Figure 1).

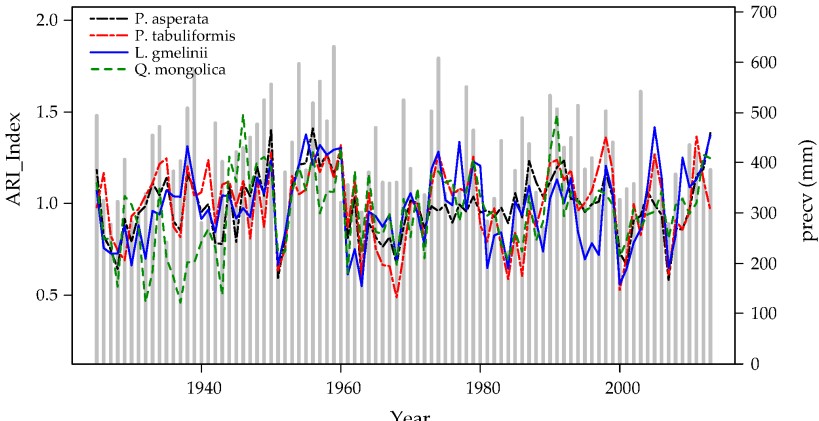

**Figure 1.** Detrended annual radial increment chronologies (ARI_Index, coloured curves) of *P. asperata* (black), *P. tabuliformis* (red), *L. gmelinii* (blue), *Q. mongolica* (green) and precipitation sum during the vegetation period (precv, grey bars) covering the 1925 to 2013 time span.

The positive factor loadings of PC2 for *Q. mongolica*, together with the respective factor scores, indicate that growth depressions during dry years are less pronounced compared to the other tree species. The factor loadings and factor scores of the four principal components as a result of the PCA are shown in Appendix B Table A2 and Figure A1.

### 3.2. Climatic Influences on ARI

All species responded significantly positively to monthly precipitation sums and significantly negatively to air temperature (Table 4). For *P. asperata* and *P. tabuliformis*, the mean monthly air temperature had a significantly negative effect on ARI in June of the current year. However, monthly precipitation sums of the previous September as well as of the current year's June had a significantly positive effect on the ARI. The same pattern for mean monthly air temperature can be seen for *Q. mongolica* where the ARI was significantly negative in June of the current year. However, for monthly precipitation sums, the result is different with a significantly negative effect only in June of the current year. For *L. gmelinii*, significant negative effects of mean monthly air temperature were observed in May and June of the current year and a significantly positive effect was present for monthly precipitation sums of the previous August. ARI was not significantly related to autumn precipitation within the year of growth ring formation. Only *Q. mongolica* responded significantly negatively to SPEI-1 in July of the current year (Figure 2).

**Table 4.** Response coefficients of monthly precipitation sums, mean monthly air temperature and SPEI–1 for the detrended ARI chronologies of the four tree species (1925–2013). Small letters represent the months of the preceding year, capital letters represent the months of the current year. N. s. = non-significant correlations.

| Climate Variable | Month | Tree Species | | | |
|---|---|---|---|---|---|
| | | *P. asperata* | *P. tabuliformis* | *L. gmelinii* | *Q. mongolica* |
| temp | MAY | n. s. | n. s. | −0.244 | n. s. |
| | JUN | −0.206 | −0.175 | −0.271 | −0.262 |
| prec | aug | n. s. | n. s. | 0.194 | n. s. |
| | sep | 0.238 | 0.186 | n. s. | n. s. |
| | MAY | n. s. | n. s. | n. s. | n. s. |
| | JUN | 0.255 | 0.242 | n. s. | 0.254 |
| SPEI-1 | JUL | n. s. | n. s. | n. s. | −0.278 |

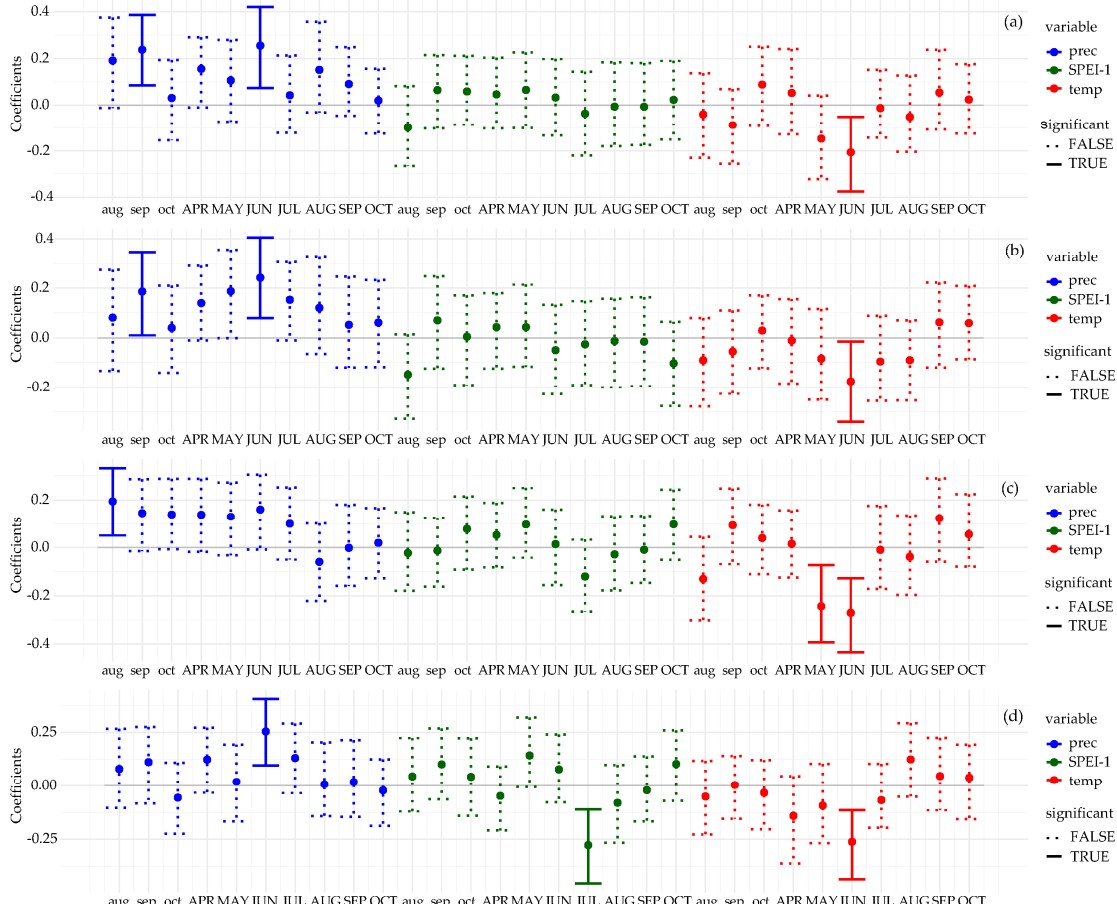

**Figure 2.** Response coefficients of mean monthly air temperature, monthly precipitation sums and SPEI-1 for the detrended ARI chronologies of *P. asperata* (**a**), *P. tabuliformis* (**b**), *L. gmelinii* (**c**) and *Q. mongolica* (**d**) (1925–2013). Small letters represent the months of the preceding year and capital letters represent the months of the current year. The darker bars indicate a coefficient significant at $p < 0.05$, the lines represent the 95%-confidence interval.

## 3.3. Temporal Stability of Dendroclimatic Relations

For the monsoon-related months of May to July, the correlation coefficients of the four detrended ARI chronologies to air temperature, precipitation and SPEI-1 are displayed in Figure 3. The climate signal in the ARI chronologies of all tree species is temporally instable. The most consistent climate

signal is a negative correlation to June air temperature and a positive correlation to June precipitation. There is no change in sign: the correlation of ARI with air temperature is negative and the correlation of ARI with precipitation positive all the time, even if the intensities of these correlations fluctuate considerably. Especially in the 1950s and 1960s, the strength of correlation changed periodically. The 1950s are characterized by mainly higher than average precipitation sums in the vegetation period and the 1940s and 1960s by precipitation sums slightly below average. In this context, the negative correlation of ARI with SPEI-1 in the dry period of 1960–1963 becomes specifically clear. The negative correlation with air temperature increases significantly in the first decade of the 21st century and decreases with precipitation. Overall, a periodically fluctuating relation with the climate parameters can be determined. The changing conditions between wet and cool as well as dry and warm periods determine the growth of all four tree species. Furthermore, in dry and warm periods the negative relation between temperature and growth is more pronounced, while in wet and cool periods the positive relation between precipitation and growth is stronger.

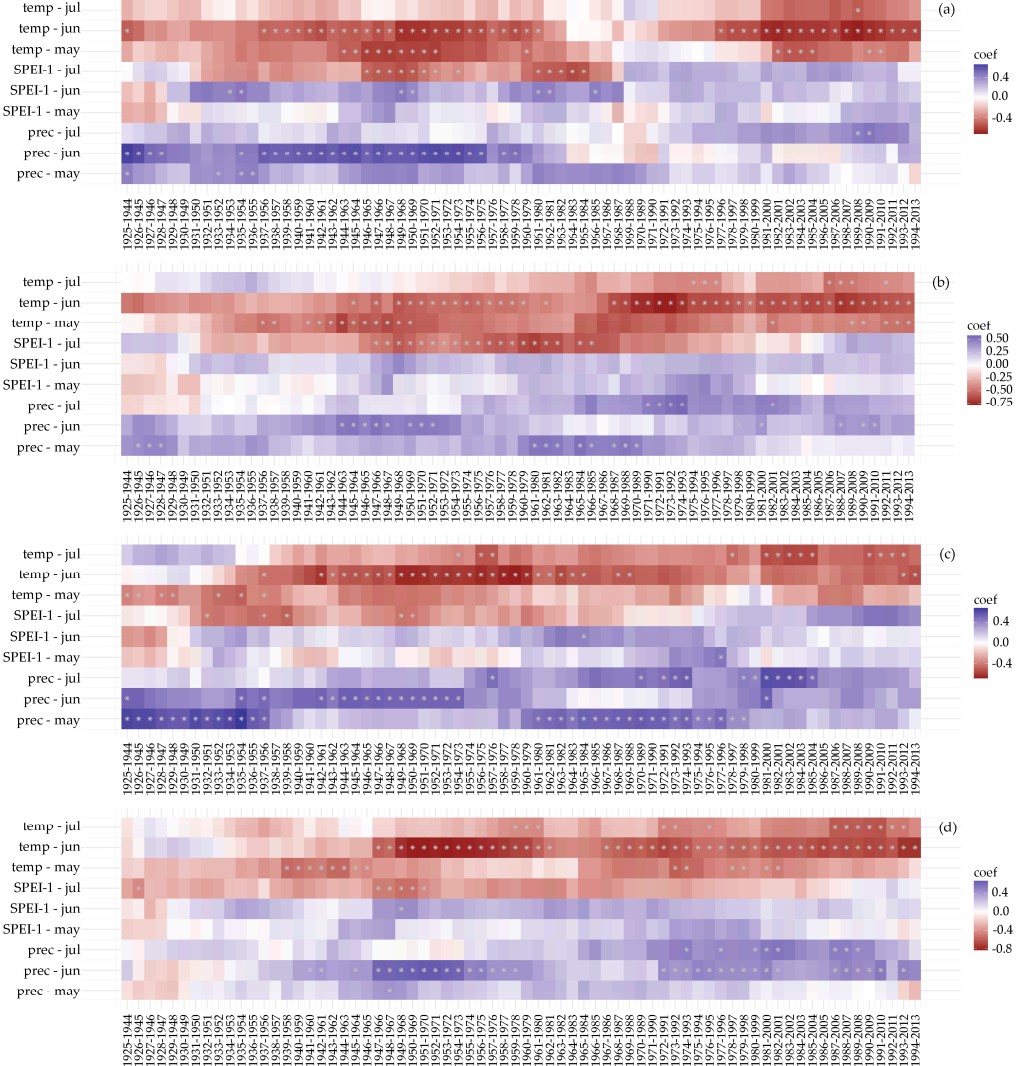

**Figure 3.** Moving correlation function relating ARI chronologies of *P. asperata* (**a**), *P. tabuliformis* (**b**), *L. gmelinii* (**c**) and *Q. mongolica* (**d**) to air temperature, precipitation and SPEI-1 of May, June and July of the current year. The moving correlation is carried out in windows of 20 years, offset by 1 year. The stars in the plot indicate windows with significant correlations for the given variable.

### 3.4. Climate Sensitivity

The aim was to examine the different tree species in terms of their climate sensitivity using an ANCOVA. The scatter of the residuals from the fitted regression line is nearly constant. Since the residuals are normally distributed and homoscedastic, the condition of using an ANCOVA is met. The diagnostic plot of the residuals of the linear mixed-effects model is shown in Appendix C Figure A2. The influence of the individual tree has been tested as random effect, as well as the variable tree species (TS). With a very low variance of 0.192, no influences of the individual tree can be demonstrated.

The effect of precv on ARI varies substantially depending on the tree species (Figure 4). The summary of the results (Table 5) shows a significant effect of precv and TSQm and also a significant interaction between precv and TSQm (precv × TSQm). The level of growth response of *Q. mongolica* to precv is significantly lower than for *P. tabuliformis*, *L. gmelinii* and *P. asperata*, indicated by lower intercept. Furthermore, the results suggest that the slope of the regression between precv and ARI is significantly different between *Q. mongolica* and the other tree species. Since *Q. mongolica* has the flattest regression line and *L. gmelinii* the steepest, *Q. mongolica* reacts less sensitively to differences in precv. Thus, the sensitivity is highest for *L. gmelinii*. The results shown in Table 5 confirm the positive relationship between ARI and precv.

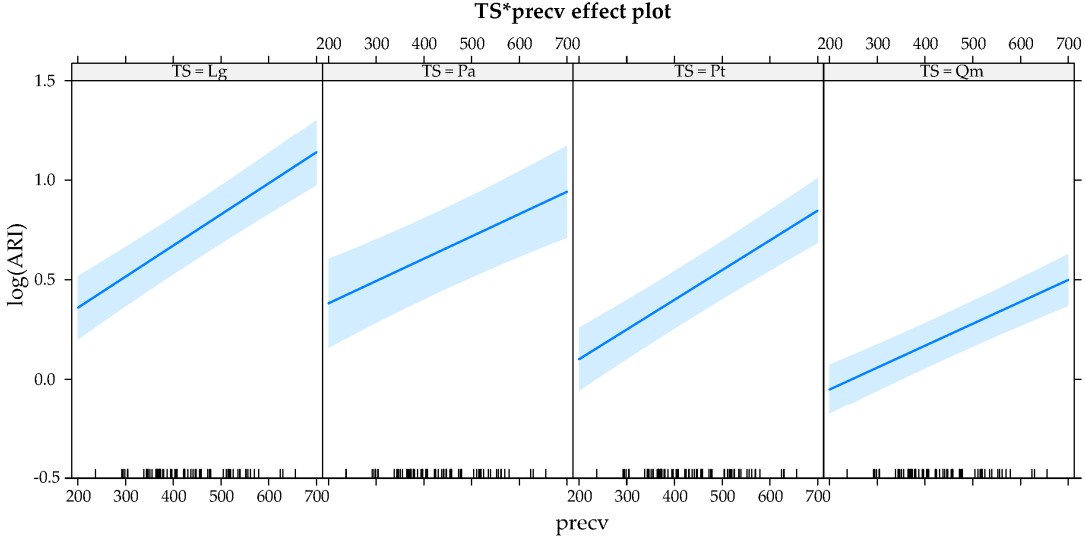

**Figure 4.** Graphical effect display for the interaction of precipitation during the vegetation period (precv) and tree species (TS). The vertical axis is labelled on the logarithm of annual radial increment (log (ARI)), and a 95-percent pointwise confidence interval is drawn around the estimated effect.

**Table 5.** ANCOVA table of the linear mixed-effects models for the tree species *P. asperata* (TSPa), *P. tabuliformis* (TSPt) and *Q. mongolica* (TSQm) with ARI as the dependent variable and TS and precv as independent variables as well as their interaction, *t*-values and *p*-values for each fixed effect. *p*-values < 0.05 $\alpha$-level are highlighted in bold type.

| Dependent Variable | Fixed Effects | Estimate | Standard Error | *t*-Value | *p*-Value |
|---|---|---|---|---|---|
|  | precv | $1.559 * 10^{-3}$ | $1.521 * 10^{-4}$ | 10.25 | **<0.001** |
|  | TSPa | $1.098 * 10^{-1}$ | $1.669 * 10^{-1}$ | 0.66 | 0.512 |
|  | TSPt | $-2.445 * 10^{-1}$ | $1.367 * 10^{-1}$ | −1.79 | 0.075 |
| ARI | TSQm | $-3.178 * 10^{-1}$ | $1.239 * 10^{-1}$ | −2.56 | **0.011** |
|  | TSPa: precv | $-4.399 * 10^{-4}$ | $2.550 * 10^{-4}$ | −1.73 | 0.084 |
|  | TSPt: precv | $-6.897 * 10^{-5}$ | $2.097 * 10^{-4}$ | −0.33 | 0.742 |
|  | TSQm: precv | $-4.608 * 10^{-4}$ | $1.924 * 10^{-4}$ | −2.40 | **0.017** |

## 4. Discussion

### 4.1. Effect of the Periodically Fluctuating Monsoon Intensities on ARI

The analysis of the four detrended ARI chronologies revealed a high synchronicity of tree-ring growth with periodically fluctuating precipitation values (Figure 1), as demonstrated by the fact that all tree species react in some individual dry years (e.g., in 1968, 1972 or 1984) with growth depression. During an extremely dry year, like 1951, trees react with strongly reduced growth, but can quickly recover under favorable weather conditions (strong monsoon) in the following year. However, the original growth level is only reached in subsequent years [10]. On the other hand, perennial drought during the summer months (e.g., 1961–1963) leads to drought stress in the vegetation period and to strong growth depression, hence trees can no longer reach their original growth level over a longer period despite favorable weather conditions in the following years [10]. Evidently, lack of precipitation during May to July of the current year and during August to October of the previous year reduces tree growth (Figure 1).

Climate in the investigated region is significantly influenced by the EASM, which causes the highest precipitation in the season of May to July, or in the case of a weak monsoon, causes low precipitation sums. The significant positive coefficients with precipitation indicate a direct effect on tree growth at strong monsoon with sufficient precipitation and the resulting cool and humid weather. Vice versa, the negative coefficients with air temperature indicate a weak monsoon, in which low precipitation sums and higher air temperatures by reduced cloud cover cause a higher evapotranspiration under warm and dry weather conditions. Therefore, the negative correlation of ARI with air temperature is not an indication of tree damage by extreme air temperatures—the maximum air temperature in this region is 21 °C (Table 1)—but rather an indication of the increased evapotranspiration with low precipitation sums at the same time. The SPEI confirms the negative influence of water deficiency on tree-ring growth in times of a weak monsoon. Liu et al. [15] reached the same conclusion and also other studies in monsoon-related areas of Northeast China support this statement [9,13,51–55]. Li et al. [9] confirm the influence of EASM for Northeast China by means of temperature reconstruction in Ningwu, based on tree-ring analysis of *P. tabuliformis*, and affirm the great consistency between various reconstructions throughout Northeast China. Recent studies exemplify further reasons for the characteristic of the macroclimate in Northeast China, such as El Niño Southern Oscillation (ENSO), Sea Surface Temperature (SST), Asian-Pacific Oscillation (APO) or Pacific Decadal Oscillation (PDO) [11,14,56]. However, for the research area EASM is identified as a crucial influencing factor of the macroclimate in the region. Further studies could focus on the effect of recurring and extreme climatic events, like El Niño and La Niña, on tree growth, since La Niña did occur in southern China in 2008 [13,57].

Deviations in analysis occur in the characteristic of influence intensity of air temperature and precipitation parameters. Bao et al. [11], Cai et al. [52], Cai et al. [53], Cai et al. [54] and Li et al. [9] found higher coefficients for the relationship of ARI with air temperature, and Gao et al. [13], Liang et al. [55] and Liu et al. [58] found higher coefficients for the relationship of ARI with precipitation. In this study, the absolute values of the positive coefficients of precipitation are marginally larger than the negative coefficients of air temperature. Slight variations in significance values of air temperature and precipitation do not allow conclusions about the extent of parameter influence.

In winter, the snow cover increases the soil temperature, which leads to an increase in root respiration, additional nutrient consumption and a negative effect on germination in the following year [59]. The strong growth response to preceding-year summer and autumn conditions emphasizes the importance of preceding-year water availability for physiologically relevant processes, such as water storage in the soil and carbohydrate storage for growth in the next year [10].

Favorable weather conditions promote the ARI and the accumulation of sufficient storage material to produce another high ARI in the following year. This storage material particularly promotes growth when the following year is characterized by a weak monsoon. Liu et al. [15] also refer to

this physiological context as substantiation for significant correlations with autumn months of the preceding year. A weak monsoon thus has the most serious impact in years with low precipitation sums in the preceding autumn when only a small amount of storage material was accumulated. Liu et al. [60] obtained similar results for radial growth of *Pinus koraiensis*.

There is no significant positive or negative relationship between the ARI and autumn precipitation of the current year. This result is consistent with the aforementioned studies. There are two reasons for this situation: firstly, the precipitation totals of these months are generally not as high as in the preceding months. Climate extremes in strong monsoon related months are no longer present, which limits the positive or negative effect on tree growth in these months. Second, ARI measurements relate to an increase in volume and the largest ARI is realized by earlywood, formed early in the vegetation period when rain and nutrients from the soil are abundant. Latewood, formed later in the season, is made of densely-layered, thick-walled cells and accounts for less tree-ring growth in terms of volume. Thus, the diverging weather conditions are prominently visible in the wider elements of earlywood than in the very small cell cavities of latewood [10,61].

## 4.2. Climate Sensitivity

Water availability is the most important and extensive factor influencing radial growth in the research area [59]. When early-summer air temperatures are high, soil moisture decreases due to limited precipitation and a high evapotranspiration rate. This limits tree growth and produces small ARI, statistically proven by the negative relationship of ARI and air temperature from May to July of the current year.

All four species showed strong seasonal drought sensitivity patterns over the whole period, but *Q. mongolica* is slightly more resistant to drought and is able to recover quicker after strong growth depression. However, in assessing the tree species, the individual level of growth has to be taken into account. With a mean ARI of 1.4 mm, the radial growth of *Q. mongolica* is considerably less than, for example, the radial growth of *P. asperata* with a mean ARI of 2.9 mm. *L. gmelinii*, *P. tabuliformis* and *P. asperata*, on the other hand, showing the most frequent growth depressions in climate records, are roughly comparable in their drought response and reduce tree-ring growth already at lower drought intensities. However, all tree species recover quickly after moderate dry years, but *L. gmelinii* reacts most sensitively to fluctuating weather conditions.

Comparing radial growth of a tree species at different latitudes may give some indications about future distribution of the species. Liu et al. [60] reported that with the rising of air temperature and precipitation remaining unchanged in the last four decades, the distribution of *P. koraiensis* would shrink. The radial growth of *P. koraiensis* decreased significantly in the southernmost point, increased significantly in the northernmost point, and did not change significantly in middle latitudes. Similar trends were also found for *L. gmelinii* [62]. For *P. tabuliformis*, rising temperatures restrain tree growth in the south but promote tree growth in the north; the distribution area may move northward [63]. However, northern tree populations may be physiologically maladapted to drought, as Isaac-Renton et al. [64] showed when investigating pine populations in Canada.

Tree radial growth variation is mainly affected by the air temperature and precipitation in summer [65]. Growth depression during summer dryness is particularly high after years with high precipitation sums in summer. Growth studies on Norway spruce in the Black Forest confirmed this effect [66]. Thus spruces reacted particularly sensitively to dry-warm summers after a series of cool-humid summers.

Besides climate and environmental factors, tree radial growth is also closely related to physiological characteristics, like photosynthesis and water conductivity [61]. Older *Larix chinensis* trees, for example, were more sensitive to climate change than younger ones, and early spring temperature and spring precipitation were the limiting factors for its growth [67].

## 5. Conclusions

The research area is located in a transitional zone between semi-arid and semi-humid conditions and is extremely sensitive and vulnerable to the intensity of the EASM. The periodically fluctuating monsoon has a strong effect on the water availability for trees, which has a vital influence on ARI in Mulan [59]. For the period of climate records in the Mulan region, the four examined ARI chronologies provide substantial results for climate–growth relations in the fluctuating monsoon climate. The most important months for tree-ring growth were the monsoon-related months May to July of the current year. Furthermore, the autumn months of the previous year, especially in times of a weak monsoon in the following year, influence the tree-ring growth. Therefore, the growth patterns of the four tree species correlate strongly with these fluctuating weather conditions. The SPEI confirms these temporal fluctuations.

In terms of climate sensitivity, *Q. mongolica* is least sensitive to differences in precipitation totals during the vegetation period. The ARI of *Q. mongolica* is significantly lower compared to the other tree species. The dominant tree species in Mulan forest, *L. gmelinii*, is most sensitive to monsoon-related climate variations. If air temperature increases and precipitation does not change, air temperature will become more relevant to the growth of *L. gmelinii*. As the degree of latitude increases, the effect of global warming on the growth of *L. gmelinii* will shift from suppression to promotion [3].

**Author Contributions:** H.S. and H.-P.K. conceived the study and contributed to its design and coordination. With the supervision of S.-M.H., all the data collection was executed by the colleagues of the Mulan Forest. Data analysis was performed by B.S., L.S., H.-P.K. and S.-M.H. under the supervision of H.S. The manuscript was written by S.-M.H. with input from B.S., L.S. and S.W. All authors contributed to the interpretation and discussion of the results. All authors have read and agreed to the published version of the manuscript.

**Funding:** This research was funded by the Federal Ministry of Education and Research (BMBF) within the project Lin2Value (Grant Number 033L049A) and by the Ministry of Science and Technology on the Chinese side (MOST, Support Code 2015DFA31440). The article processing charge was funded by the Baden-Württemberg Ministry of Science, Research and Art and the University of Freiburg in the funding program Open Access Publishing.

**Acknowledgments:** All climate data was provided by the Chinese Academy of Forestry in Beijing.

**Conflicts of Interest:** The authors declare no conflict of interest. The funders had no role in the design of the study; in the collection, analyses, or interpretation of data; in the writing of the manuscript, or in the decision to publish the results.

## Appendix A

**Table A1.** Levene's Test for homogeneity of variance for the gridded datasets and the data of Weichang meteorological station for the period 1951–2013. SD = Standard deviation. There is a significant relation if $p > 0.05$ $\alpha$-level.

| Months | Precipitation (mm) | | | Air Temperature (°C) | | |
|---|---|---|---|---|---|---|
| | *p*-Value | SD Weichang | SD Gridded Data | *p*-Value | SD Weichang | SD Gridded Data |
| Jan | 0.817 | 2.3 | 2.3 | 0.750 | 1.8 | 1.7 |
| Feb | 0.111 | 2.8 | 2.2 | 0.481 | 2.6 | 2.4 |
| Mar | 0.948 | 6.7 | 6.4 | 0.577 | 2.0 | 2.1 |
| Apr | 0.115 | 15.6 | 10.6 | 0.592 | 1.7 | 1.5 |
| May | 0.132 | 23.9 | 17.2 | 0.621 | 1.0 | 1.1 |
| Jun | 0.073 | 35.0 | 26.6 | 0.942 | 1.0 | 1.0 |
| Jul | 0.060 | 54.0 | 42.7 | 0.739 | 1.1 | 1.0 |
| Aug | 0.069 | 45.1 | 31.5 | 0.816 | 0.9 | 0.9 |
| Sep | 0.102 | 29.7 | 18.5 | 0.474 | 1.0 | 1.1 |
| Oct | 0.418 | 15.2 | 12.6 | 0.695 | 1.2 | 1.2 |
| Nov | 0.814 | 5.2 | 5.0 | 0.736 | 1.7 | 1.6 |
| Dec | 0.921 | 1.8 | 1.8 | 0.479 | 2.1 | 1.9 |

## Appendix B

**Table A2.** Summary statistics of the Principal Components Analysis (PCA)-matrix of factor loadings. Variable = detrended ARI chronologies of the four tree species.

| Variable | Observation | Factor Loadings of the Principal Components | | | |
|---|---|---|---|---|---|
| | | PC 1 | PC 2 | PC 3 | PC 4 |
| *L. gmelinii* | | 7.928 | −2.192 | 4.121 | 1.831 |
| *P. tabuliformis* | Calendar years | 8.442 | −0.771 | 0.051 | −4.018 |
| *P. asperata* | (1925–2013) | 7.879 | −2.508 | −4.060 | 1.774 |
| *Q. mongolica* | | 6.640 | 6.574 | −0.168 | 0.816 |
| Variance (%) | | 68.3 | 15.6 | 9.5 | 6.6 |

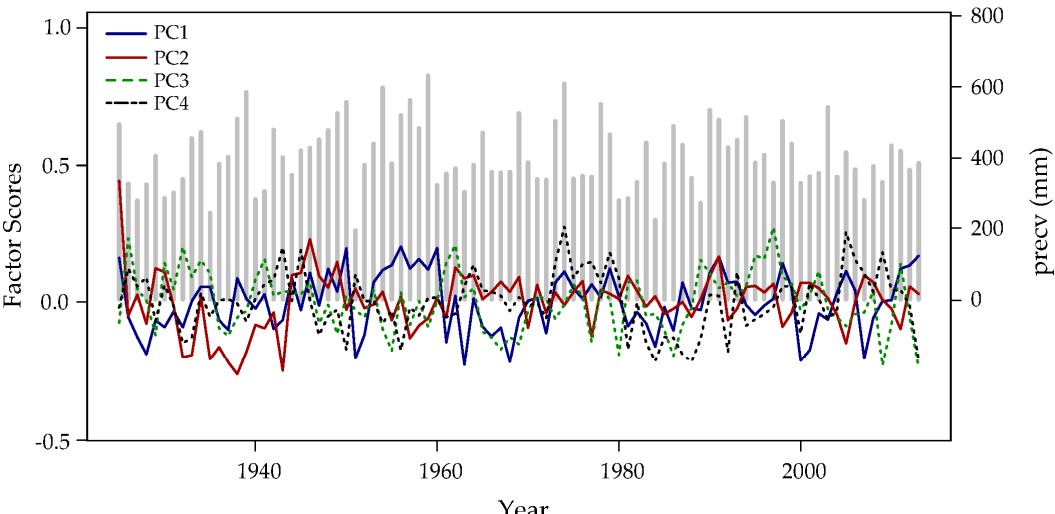

**Figure A1.** Principal components analysis (PCA): Factor Scores of the four principal components (coloured lines) and precipitation sum during the vegetation period (precv, bars) covering the 1925 to 2013 time span.

## Appendix C

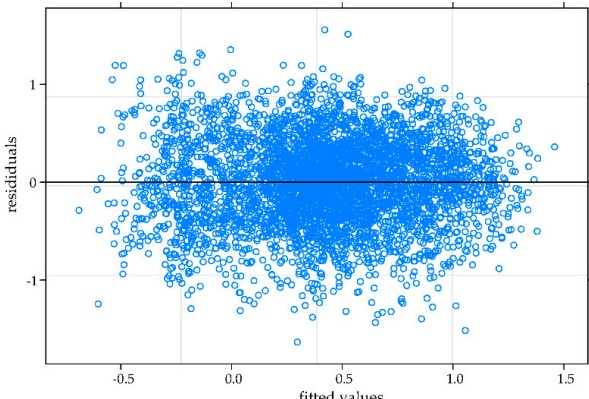

**Figure A2.** Scatter plot of the model residuals versus fitted values of the linear mixed-effects model for the four tree species.

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
