# Peer review of "Impact of Precipitation and Temperature Variability of the East Asian Summer Monsoon (EASM) on Annual Radial Increment of Selected Tree Species in Northeast China"

_forests, doi:10.3390/f11101093_

Round 1

Reviewer 1 Report

This manuscript describes an effort to relate the annual growth of four species of trees to climate conditions which are greatly influenced by periodic monsoon events. The approach compares annual tree ring growth data to climate data for the study area through the application of several analytical methods that are commonly applied to time series data, particularly tree ring series. The results presented indicate a qualitatively similar response of the four tree species to climate drivers, with a positive response to precipitation in specific periods from both the current and prior year--this is a common outcome of climate response analysis as trees cannot capitalize on incoming precipitation at all points of a year in temperate and boreal biomes but can access these inputs at later points through deep roots. The results also indicate that the deciduous species of the genus Quercus was less sensitive to precipitation variation than co-occuring conifers, which also agrees with data from other locations where oaks are able to better tolerate climate fluctuations and warming conditions relative to native conifers.

Overall, the paper is reasonably well-written with a few sections that would benefit from either being moved to an alternate location in the manuscript, or edited slightly to improve clarity. The results, while not novel in the larger sense of dendrology, are a valuable contribution to the expanding body of literature on tree responses to climate and are sure to be of interest to others in this field. 

Major Comments:

Table 2: If I understand the sample design, stands are independent of each other based on distance. So it seems the original goal was to collect 3 trees from each of 10 stands for a total of 30 trees per species. Is this correct? If so, it seems that your analyses should include a term for "stand ID" to partition variance at the stand level.

Line 177: Unfortunately, I am not able to read German and therefore cannot access the methodology described here for extrapolating climate data. How this extrapolation was completed could be critical to understanding the relationship of tree growth and climate prior to the period when local measures were collected. Please expand on this method in brief. My primary concern is that the extrapolation approach uses either climate data from some location that is far removed from the study site, or that it draws upon tree growth data in the reconstruction which would violate an assumption of independence among the calculation of the predictor and response variable.

The Discussion section begins with information that is better suited to the Methods or results section of the paper. A discussion of the results of this study really begins on line 390; all the information prior to this point is important for contextualizing the analyses but distracts from the discussion. I would like to see the discussion presented in two sections that related back to the two hypotheses presented in the introduction. This approach would better frame the study and help to avoid the distraction presented by the inclusion of methods.

Minor Comments:

Line 88: Please make this past tense: "The hypotheses tested were:" 

Line 129 - 130: Please make this past tense: "To be included in our study, sample trees were required to have a healthy crown and no deterioration..."

Line 295 - 296: consider rephrasing to "ARI was not significantly related to autumn precipitation within the year of growth ring formation." Or something similar to this, as currently written, the sentence is a bit clumsy.

Reviewer 2 Report

Dear authors,

First, I would like to thank you for presenting your results. Overall, I found that the work is very interesting.  In my opinion the research was done very well.  The study design setup and analysis are very good, with sufficient sample size. The article is understandably written and well-organized, contain all the components I would expect, and the sections are well-developed. The methodology is clearly explained, the results are well described, and the discussion carried out very well. Good and sufficient bibliography allows the readers with a less knowledge of dendrochronology to get to a lot of information. In my opinion, very good paper. 

However, I have few comments and suggestions:

  1. I would suggest adding or moving the following information from the section Results to the section Materials and methods

   Line296-297 ...... ” For all results of the bootstrapped response function analysis, the α-level was set to 0.05”.

  1. In the section Materials and methods, Time stability of dendroclimatic relations, I would suggest adding information about the window of moving correlation. You give it in Figure 5, (line 329) but it would be worth including in the methodology.
  1. It would be advisable to standardize the method of citation, in my opinion it is correct to put the author's name at the beginning of the sentence and the reference number in brackets. However, in other cases, a uniform citation is indicated, i.e. the reference number in brackets parentheses. - please review the introduction and discussion in this regard (e.g. lines: 74, 475).

Best regards,
